# Computational prediction of the pathogenic variants of arachidonate 5-lipoxygenase activating protein using Molecular Dynamics simulation

Mohamed E. Elnageeb[1], Imadeldin Elfaki[2]*, Gad Allah Modawe[3],
Abdelrahman Osman Elfaki[4], Othman R. Alzahrani[5], Hytham A. Abuagla[1],
Hayam A. Alwabsi[2], Adel I. Alalawy[2], Mohammad Rehan Ajmal[2],
Elsiddig Idriss Mohamed[6], Hussein Eledum[6], Syed Khalid Mustafa[7],
Elham M. Alhathli[8]

1 Department of Medical Laboratory Sciences, College of Applied Medical Sciences, University of Bisha, Bisha, Kingdom of Saudi Arabia, 2 Department of Biochemistry, Faculty of Science, University of Tabuk, Tabuk, Kingdom of Saudi Arabia, 3 Department of Biochemistry, Faculty of Medicine and Heath Science, Omdurman Islamic university, Omdurman, Sudan, 4 Department of Computer Science, Faculty of Computers and Information Technology, University of Tabuk, 5 Department of Biology, Faculty of Sciences, University of Tabuk, Tabuk, Kingdom of Saudi Arabia, 6 Department of Statistics, Faculty of Science, University of Tabuk, Tabuk, Kingdom of Saudi Arabia, 7 Department of Chemistry, Faculty of Science, University of Tabuk, Tabuk, Kingdom of Saudi Arabia, 8 Faculty of Nursing, Taif University, Taif, Kingdom of Saudi Arabia

* elfakiimadeldin@gmail.com

## Abstracts

The arachidonate 5-lipoxygenase activating protein (ALOX5AP) regulates leukotrienes (LTs) synthesis. LTs are involved in inflammation which is implicated in cardiovascular diseases (CVDs) and stroke. Variations in *ALOX5AP* gene are associated with CVDs, stroke and others because of their possible effects on ALOX5AP stability and function. In this study we investigated with molecular dynamics (MD) simulation the structural impacts of L12F, A56V, G75R, and G87R variants on ALOX5AP. We employed an array of bioinformatics techniques, including SIFT, PolyPhen-2, PANTHER, SNPs&GO, PhD-SNP, i-Mutant, MuPro, MutPred, ConSurf, and GROMACS. Results showed that the L12F variant increased structural compactness, as indicated by diminished solvent accessibility, a reduced radius of gyration, and a decrease in hydrogen bonding capacity. The A56V variant destabilized the ALOX5AP, demonstrating elevated root mean square deviation (RMSD), augmented solvent-accessible surface area, and diminished ALOX5AP compactness. The G75R and G87R variants exhibited mild effects on ALOX5AP wildtype. However, simulation trajectory snapshots results indicated G75R and G87R variants induce instability leading to structural perturbations of ALOX5AP probably due to the charge of arginine introduced by the G75R and G87R mutation. The G75R and G87R variants potentially influence ALOX5AP dynamics, stability, and function. These results require further verification in future case-control and protein functional studies.

**Data availability statement:** All relevant data are within the manuscript and its Supporting information files.

**Funding:** The author(s) received no specific funding for this work.

**Competing interests:** The authors have declared that no competing interests exist

**Abbreviations:** ALOX5AP, Arachidonate 5-Lipoxygenase Activating Protein; FLAP, 5-Lipoxygenase Activating Protein; 5-LO, 5-Lipoxygenase; LT, Leukotrienes; LTA4, Leukotriene A4; LTB4, Leukotriene B4; LTC4, Leukotriene C4; LTD4, Leukotriene D4; LTE4, Leukotriene E4; LTC4S, Leukotriene C4 Synthase; CVDs, Cardiovascular Diseases; CAD, Coronary Artery Disease; MI, Myocardial Infarction; SNP, Single Nucleotide Polymorphism; nsSNP, Non-Synonymous Single Nucleotide Polymorphism; MD, Molecular Dynamics; RMSD, Root Mean Square Deviation; RMSF, Root Mean Square Fluctuation; Rg, Radius of Gyration; SASA, Solvent Accessible Surface Area; HB, Hydrogen Bond; PDB, Protein Data Bank; NCBI, National Center for Biotechnology Information; OMIM, Online Mendelian Inheritance in Man

## 1. Introduction

Cardiovascular diseases (CVDs) are important cause of premature death and morbidity all over the world with increasing rate of incidence [1]. Atherosclerosis is the underlying cause of pathogenesis and progression of CVDs including coronary artery disease (CAD) or ischemic heart disease, cerebrovascular disease or stroke, venous thromboembolism and, peripheral vascular disease [1]. Inflammation is one of the important pathways in the formation of atherosclerosis resulting in the atherosclerotic plaque and the subsequent pathologies such as CAD and stroke [2,3]. CVDs are developed through the interactions of genetic and environmental risk factors [4]. Environmental risk factors include unhealthy diet, physical inactivity, smoking, male gender, obesity, dyslipidemia, and air pollution [5]. The genetic risk factors were identified through the genome-wide association studies that revealed the linkage of certain loci with diseases such as diabetes mellitus (DM), CVDs, and cancers [6,7].

Arachidonate 5-Lipoxygenase Activating Protein (ALOX5AP) is encoded by *ALOX5AP* gene, and also known as Human 5-Lipoxygenase Activating Protein (FLAP). ALOX5AP is important for 5-lipoxygenase pathway (5-LO), this pathway is required for leukotrienes biosynthesis [8,9]. The human *ALOX5AP* gene is found on chromosome 13q12-13 and composed of 5 exons, 4 introns [10]. Leukotrienes are derived from arachidonic acid and function as inflammation mediators [11]. The biosynthesis of leukotrienes starts with the cleavage of arachidonic acid from the membrane phospholipids [12,13]. The 5-Lipoxygenase (5-LO), activated by FLAP, and catalyzes the conversion of arachidonic acid to 5-hydroperoxyeicosatetraenoic acid and then to leukotriene A4 (LTA4) [13]. The LTA4 is either converted to leukotriene B4 (LTB4) by LTA4 hydrolase, or conjugated to glutathione by LTC4 synthase (LTC4S) generating the cysteinyl leukotrienes such as leukotriene C4 (LTC4), leukotriene D4 (LTD4), and leukotriene E4 (LTE4) [13]. Leukotrienes are involved in autoimmune and inflammatory, CVDs, and tumors [14]. It has been reported that the 5-LO pathway is upregulated in cardiovascular diseases and that cysteinyl leukotrienes are implicated in atherosclerosis, CAD and stroke [13]. In addition the *ALOX5AP* gene variations are associated with CAD risk in patients with familial hypercholesterolemia [15], stroke in Iranian, Chinese [16,17], and premature CAD in European American patients [18]. However, some previous studies indicated no association of certain *ALOX5AP* gene polymorphisms with stroke in Chinese [19] and myocardial infarction (MI) in European population [20]. Moreover, the *ALOX5AP* gene polymorphisms were reported to be associated with thyroid cancer [21], myeloid leukemia [22] and Alzheimer's disease [23]. In this research, we examined the effects of the *ALOX5AP* gene variations L12F, A56V, G75R, and G87R on ALOX5AP with molecular dynamic simulation [24–27] as ALOX5AP variants may be associated with CVDs, and other diseases.

## 2. Methods

### 2.1. Plan of work

The present study employed computational methodologies and various bioinformatics technologies [24,27–30] to examine the effects of non-synonymous single nucleotide

polymorphisms (nsSNPs) in the ALOX5AP. Then the nsSNPs with a high-risk profile were chosen for additional examination, involving an evaluation of their conservation, stability, and structural impact using molecular dynamics (MD) simulation (Fig 1).

## 2.2. Data collection

The nucleotide and amino acid sequences of the ALOX5AP, accession numbers NG_011963.2 and NP_001620.2, were obtained from the NCBI ((accessed on 23 May 2024) in FASTA format. The SNP data for the *ALOX5AP* gene is accessible on the NCBI SNP database at the following URL: http://www.ncbi.nlm.nih.gov/snp/ accessed on 23 May 2024. Additionally, relevant information concerning the *ALOX5AP* gene and protein was obtained from the PDB ID 2Q7M, the Uniprot database (https://www.uniprot.org/uniprotkb/P20292/entry) accessed on 23 May 2024, and the Online Mendelian Inheritance in Man (OMIM) database, available at http://www.omim.org (accessed on 23 May 2024).

## 2.3. Prediction of functional impact of nsSNPs

This study does not include any human or animal subjects performed by any coauthors and therefore ethical approval and consent were not required. To identify deleterious nsSNPs in the *ALOX5AP* gene, a multi-tool computational approach was employed, utilizing predictive algorithms and structural analyses in the following steps. We examined the impact of genetic variations on protein function with numerous online tools and servers, including SIFT [31,32] which assesses the degree of conservation of amino acids and the effects of substitutions, categorizing variants as "tolerated" or "deleterious. In SIFT score, the amino acid substitution is predicted damaging if the score is <= 0.05 [32]. PolyPhen2 [33] predicts the functional consequences based on sequence and structural features, classifying variants as "benign," "possibly damaging," or "probably damaging.", and PANTHER [34] that was used to classify variants based on evolutionary conservation and biological significance, identifying those with potential deleterious effects.

## 2.4. Structural stability and functional impact assessment

The tools employed for evaluating the correlation between the filtered SNPs and disease were PhD-SNP [35,36] to provide support vector machines and evolutionary information and SNPS&GO [37] to evaluate the functional impact using neural

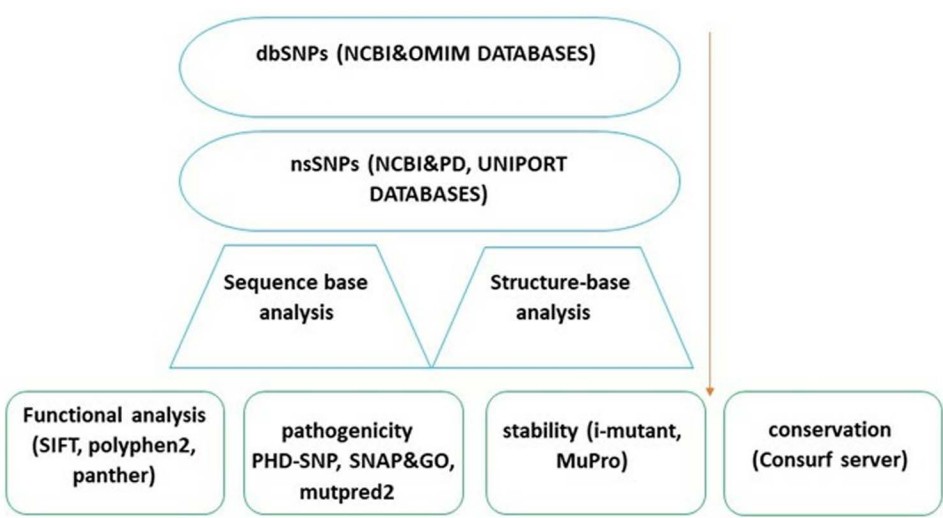

**Fig 1. Schematic representation of the workflow.**

networks. The I-Mutant algorithm receives the amino acid sequence of the ALOX5AP, together with data on the individual mutated residues and their corresponding locations [38]. The MUpro tool was utilized to assess protein stability and changes in mutations [39]. While the MutPred2 web service was used for classifying mutations as either neutral or disease-associated [40], and the assessment of amino acid conservation was performed using the ConSurf web server [41].

## 2.5. Molecular Dynamics (MD) simulations using GROMACS

The simulation workflow involved multiple steps, including system preparation, energy minimization, equilibration, production simulation, and subsequent post-simulation analysis to evaluate structural and dynamic properties were performed using GROMACS [42]. The initial structures were prepared using CHARMM-GUI, specifically its Solution Builder module, which facilitates system setup with CHARMM36m forcefield. This process ensured accurate representation of the molecular system through solvation, ion addition, and parameterization. To refine the system further, energy minimization was carried out to eliminate steric clashes and stabilize the structure. Equilibration was performed in two phases to enable the system to adapt to the desired simulation conditions while maintaining structural integrity. Beginning with the previously minimized structure, equilibration established a well-stabilized environment for the production run. Following this, production simulations were conducted over a time scale of 100 nanoseconds (ns), encompassing 50 million steps. These simulations leveraged GPU acceleration to enhance computational efficiency, with specific commands such as `-bonded gpu` and `-pme gpu` employed to optimize the handling of bonded and non-bonded interactions. The production phase generated detailed atomistic trajectories, providing insight into the dynamic evolution of the protein structures. Post-simulation analysis was undertaken using GROMACS [42] to examine the structural stability and flexibility of ALOX5AP key metrics such as the root-mean-square deviation (RMSD) [29,43], which assesses the ALOX5AP overall structural stability, and root-mean-square fluctuation (RMSF) [29,43], which evaluated the flexibility of individual amino acid residue. We also examined the radius of gyration (Rg) which gives an understanding of the overall dimensions of ALOX5AP, and solvent accessible surface area (SASA) that is a bio-molecular surface area accessible to solvent molecules [29,44]. To enhance the data interpretation, custom Python scripts were developed [45] to generate visualizations of these parameters, enabling a comprehensive understanding of the simulation results. Hydrogen bonds (HB) formation analysis was also examined, HB analysis is an important factor in protein stability, structural integrity, and interactions [46]. Furthermore, trajectory snapshots per 100th frame of the entire simulation period were taken as protein data bank (PDB) files which were then superimposed to compare the structural changes of the variants structure overtime.

## 3. Results

### 3.1. Prediction of deleterious nsSNPs in *ALOX5AP*

A thorough computational investigation was conducted to discover harmful nsSNPs in the *ALOX5AP* gene utilizing SIFT, PolyPhen-2, and PANTHER tools. The findings indicated several variations with a significant likelihood of being harmful, corroborated by uniform predictions from all three methods. SIFT classified all examined nsSNPs as harmful, with scores between 0 and 0.03. The scores demonstrate significant evolutionary conservation at the impacted residues, implying that alterations at these locations are likely to impair protein function. Significant variations with a score of 0 encompass L12F, G46R, and G75R, underscoring their potential influence on protein stability and functionality. PolyPhen-2 categorized all variations as "probably damaging," with scores approaching or equal to 1, signifying a substantial probability of functional impairment. Variants including I18N, G46R, and P65R received a maximum score of 1, thereby reinforcing their anticipated deleterious impacts. These elevated scores indicate substantial modifications in protein structure or interactions resulting from these mutations. PANTHER categorized the majority of the variations as "probably damaging," with Pdel values between 0.57 and 0.95. Variants including (F50L), (N59K), G87R, and G100S

had the highest Pdel values of 0.95, signifying a substantial probability of adverse impacts grounded in evolutionary conservation and functional significance. A number of nsSNPs were consistently predicted as harmful by all techniques. Notable variants, such as L12F, G46R, G75R, and G87R, exhibited elevated scores in SIFT, PolyPhen-2, and PANTHER (Table 1), suggesting their considerable capacity to influence the structure and function of the ALOX5AP. These findings establish a solid foundation for additional structural and functional investigations to elucidate the significance of these mutations in the etiology of CVDs.

**Table 1. SIFT, PolyPhen, and PANTHER predictions for the impact of amino acid substitution on the *ALOX5AP* gene.**

| Variant ID | Alleles | AA | SIFT | | Polyphen | | PANTHER | |
| --- | --- | --- | --- | --- | --- | --- | --- | --- |
| | | | Predication | Score | Predication | Score | Message | Pdel |
| rs764814001 | G/T | D2Y | Deleterious | 0.03 | probably damaging | 0.999 | probably damaging | 0.57 |
| rs199916092 | G/T | L12F | Deleterious | 0 | probably damaging | 0.997 | probably damaging | 0.78 |
| rs775787793 | T/A | I18N | Deleterious | 0.01 | probably damaging | 1 | probably damaging | 0.57 |
| rs775787793 | T/G | I18S | Deleterious | 0.01 | probably damaging | 1 | probably damaging | 0.57 |
| rs1278903858 | C/A | A27D | Deleterious | 0.02 | probably damaging | 0.999 | probably damaging | 0.57 |
| rs1278903858 | C/T | A27V | Deleterious | 0.03 | probably damaging | 0.998 | probably damaging | 0.57 |
| rs41351946 | C/G | S41R. | Deleterious | 0.02 | probably damaging | 0.994 | probably damaging | 0.57 |
| rs1438548272 | G/A | G46R | Deleterious | 0 | probably damaging | 1 | probably damaging | 0.85 |
| rs1302091419 | T/G | F50L | Deleterious | 0 | probably damaging | 0.992 | probably damaging | 0.95 |
| rs748246562 | A/G | Y54C | Deleterious | 0.01 | probably damaging | 1 | probably damaging | 0.57 |
| rs781044231 | C/T | A56V | Deleterious | 0 | probably damaging | 0.998 | probably damaging | 0.86 |
| rs777940375 | C/A | N59K | Deleterious | 0 | probably damaging | 0.998 | probably damaging | 0.95 |
| rs563599872 | G/T | C60F | Deleterious | 0 | probably damaging | 0.997 | probably damaging | 0.78 |
| rs1245911587 | A/T | D62V | Deleterious | 0.01 | probably damaging | 0.998 | probably damaging | 0.57 |
| rs768483394 | C/G | P65R | Deleterious | 0 | probably damaging | 1 | probably damaging | 0.95 |
| rs1338792287 | C/G | L71V | Deleterious | 0 | probably damaging | 0.997 | probably damaging | 0.57 |
| rs1338792287 | C/T | L71F | Deleterious | 0 | probably damaging | 0.962 | probably damaging | 0.57 |
| rs148308449 | C/A | A74E | Deleterious | 0 | probably damaging | 0.999 | probably damaging | 0.78 |
| rs148308449 | C/T | A74V | Deleterious | 0 | probably damaging | 1 | probably damaging | 0.78 |
| rs376956587 | G/A | G75R | Deleterious | 0 | probably damaging | 1 | probably damaging | 0.78 |
| rs1951856934 | T/C | L76P | Deleterious | 0.03 | probably damaging | 1 | probably damaging | 0.57 |
| rs1951890039 | C/T | A83V | Deleterious | 0 | probably damaging | 0.999 | probably damaging | 0.78 |
| rs1951890190 | G/A | G87R | Deleterious | 0 | probably damaging | 1 | probably damaging | 0.95 |
| rs1489183133 | G/T | R94S | Deleterious | 0 | probably damaging | 1 | probably damaging | 0.95 |
| rs1383209302 | G/A | G100S | Deleterious | 0 | probably damaging | 1 | probably damaging | 0.95 |
| rs1383209302 | G/C | G100R | Deleterious | 0 | probably damaging | 1 | probably damaging | 0.95 |
| rs1345574893 | T/G | I119R | Deleterious | 0.01 | probably damaging | 0.998 | probably damaging | 0.57 |
| rs200791383 | C/A | L122M | Deleterious | 0 | probably damaging | 1 | probably damaging | 0.57 |
| rs984005136 | T/G | L122R | Deleterious | 0 | probably damaging | 0.999 | probably damaging | 0.57 |
| rs750529523 | T/C | M125T | Deleterious | 0.01 | probably damaging | 0.997 | probably damaging | 0.57 |
| rs750529523 | T/G | M125R | Deleterious | 0.01 | probably damaging | 0.967 | probably damaging | 0.57 |
| rs751354742 | T/A | Y134N | Deleterious | 0 | probably damaging | 0.997 | probably damaging | 0.57 |
| rs751354742 | T/C | Y134H | Deleterious | 0.01 | probably damaging | 0.991 | probably damaging | 0.57 |
| rs1951969673 | C/A | S155Y | Deleterious | 0 | probably damaging | 0.999 | probably damaging | 0.57 |
| rs774719216 | C/T | P161S | Deleterious | 0 | probably damaging | 1 | probably damaging | 0.57 |

## 3.2. Prediction of pathogenic nsSNPs in ALOX5AP using SNPs&GO and PhD-SNP

To assess the possible pathogenicity of nsSNPs in the *ALOX5AP* gene, the SNPs&GO and PhD-SNP tools were utilized. Both techniques reliably identified significant variations as disease-associated with high confidence. SNPs&GO categorized all four examined nsSNPs as "pathogenic," with pathogenicity probability (Path_Prop) between 0.584 and 0.953 [37]. The variations G75R and G87R had the greatest pathogenicity probability of 0.953 each, signifying a robust correlation with functional impairment. The mutation L12F exhibited a significant likelihood of pathogenicity (0.860), hence reinforcing its harmful characteristics. The mutation A56V, despite a comparatively lower Path_Prop of 0.584, was nonetheless categorized as harmful, indicating possible functional implications. PhD-SNP classified all four variations as "disease-associated," with reliability indices (RI) between 2 and 9, indicating the confidence in the predictions [36]. Variants G75R and G87R received the highest RI values of 9, suggesting a significant probability of becoming pathogenic variants. The variant L12F was firmly identified as disease-associated, whereas A56V obtained a risk index score of 2, indicating a moderate probability of disease association. The uniform designation of these nsSNPs as harmful by both SNPs&GO and PhD-SNP underscores their potential influence on the structure and function of the ALOX5AP. Significant variants, such as G75R and G87R were identified as the most detrimental alterations, corroborated by elevated pathogenicity probability and reliability scores (Table 2).

## 3.3. Prediction of ALOX5AP stability changes using i-Mutant and MuPro

The impact of specific non-synonymous single nucleotide polymorphisms (nsSNPs) in the ALOX5AP on structural stability was assessed utilizing i-Mutant and MuPro (Table 3). Both instruments yielded uniform forecasts on the destabilizing effects of particular variants.

i-Mutant forecasted a reduction in protein stability for three of the examined variations, namely L12F, G75R, and G87R. The destabilizing effects were corroborated by stability scores of 3, 6, and 1, respectively. The variant A56V was predicted to enhance stability, with a score of 6, suggesting a possible stabilizing influence on the protein structure. MuPro forecasted that all four variations will diminish ALOX5AP stability, specifically L12F, A56V, G75R, and G87R. The predictions aligned with i-Mutant for the destabilizing variants and offered a different viewpoint for the stabilizing prediction of A56V. i-Mutant and MuPro consistently recognized L12F, G75R, and G87R as destabilizing mutations, indicating their potential

Table 2. Analysis of disease-associated nsSNPs on *ALOX5AP* gene.

| Variant ID | Alleles | Mutation | SNPs&GO | | | PhD-SNP | |
| --- | --- | --- | --- | --- | --- | --- | --- |
| | | | Pred_class | Path_Prop | RI | Prediction | Score |
| rs199916092 | G/T | L12F | Pathogenic | 0.85970382 | 7 | Disease | 0 |
| rs781044231 | C/T | A56V | Pathogenic | 0.5844504 | 2 | Disease | 1 |
| rs376956587 | G/A | G75R | Pathogenic | 0.95299451 | 9 | Disease | 4 |
| rs1951890190 | G/A | G87R | Pathogenic | 0.95299451 | 9 | Disease | 3 |

Table 3. Investigation of the molecular mechanisms underlying pathogenicity.

| Variant ID | Alleles | AA | i-mutant | | MuPro |
| --- | --- | --- | --- | --- | --- |
| | | | Prediction | score | prediction |
| rs199916092 | G/T | L12F | decrease | 3 | decrease |
| rs781044231 | C/T | A56V | increase | 6 | decrease |
| rs376956587 | G/A | G75R | decrease | 6 | decrease |
| rs1951890190 | G/A | G87R | decrease | 1 | decrease |

effect on the structural integrity and functionality of the ALOX5AP. The variant A56V displayed inconsistent predictions, necessitating more examination via MD simulations.

### 3.4. Prediction of functional impacts of nsSNPs using MutPred

All four examined nsSNPs received elevated pathogenicity scores, varying from 0.766 to 0.933 (Table 4). The anticipated functional effects are as follows: The L12F variant had a pathogenicity score of 0.766 and was linked to several functional modifications, such as the loss of helix and loop structures, absence of GPI-anchor amidation at N8, and disturbances in the transmembrane protein domain and signal peptide. The A56V scored 0.809 and may in the loss of a helical shape, potentially compromising local protein folding and stability. While the G75R variant received a score of 0.859 and is linked to the loss of pyrrolidone carboxylic acid at Q80, an alteration that could influence protein stability or interactions. The G87R variant demonstrated the greatest pathogenicity score of 0.933 and was anticipated to modify an ordered interface, perhaps disrupting protein-protein interactions or binding affinity. The elevated pathogenicity scores and anticipated functional effects underscore the substantial influence of these nsSNPs on the structural and functional characteristics of the ALOX5AP. The variations G75R and G87R are significant due to their elevated scores and the anticipated disruption of essential functional characteristics, including post-translational modifications and protein interfaces.

### 3.5. Conservation analysis of ALOX5AP using ConSurf

The evolutionary conservation of amino acid residues in the ALOX5AP was assessed using ConSurf. The research indicated differing degrees of conservation throughout the protein sequence, with certain residues recognized as highly conserved and possibly essential for protein function. The results of the conservation study for the four mutants L12, A56, G75, and G87, derived from the ConSurf service, is illustrated in Fig 2. The findings indicate that these mutants had a high conservation score of 8 or 9, signifying that these residues are highly conserved and situated inside essential structural domains of the ALOX5AP. These findings indicate that mutations at these loci may exert considerable functional or structural effects. Conversely, residues with lower conservation scores, predominantly situated in more changeable and exposed areas of the protein, were less probable to be crucial for preserving the protein's basic function. The significant conservation of L12, A50, G75, and G87 underscores their probable significance in the stability and functionality of the protein.

### 3.6. Molecular dynamics (MD) simulation of ALOX5AP wildtype and its variants

**3.6.1. Temperature, pressure, and density analyses of ALOX5AP wildtype and its variants.** The temperature, pressure, and density characteristics of the wildtype ALOX5AP and its variants (L12F, A56V, G75R, and G87R) were examined throughout a 100 ps simulation period to evaluate the structural and dynamic impacts of the mutations.

**Table 4. Investigation of the molecular mechanisms underlying pathogenicity.**

| Variant ID | Alleles | AA | Mut-pred | | Functions affected |
| --- | --- | --- | --- | --- | --- |
| | | | score | Prediction | |
| rs199916092 | G/T | L12F | 0.766 | -ve | Loss of Helix; Loss of Loop; Loss of GPI-anchor amidation at N8; Altered Transmembrane protein; Altered Signal peptide |
| rs781044231 | C/T | A56V | 0.809 | -ve | Loss of Helix |
| rs376956587 | G/A | G75R | 0.859 | -ve | Loss of Pyrrolidone carboxylic acid at Q80 |
| rs1951890190 | G/A | G87R | 0.933 | -ve | Altered Ordered interface |

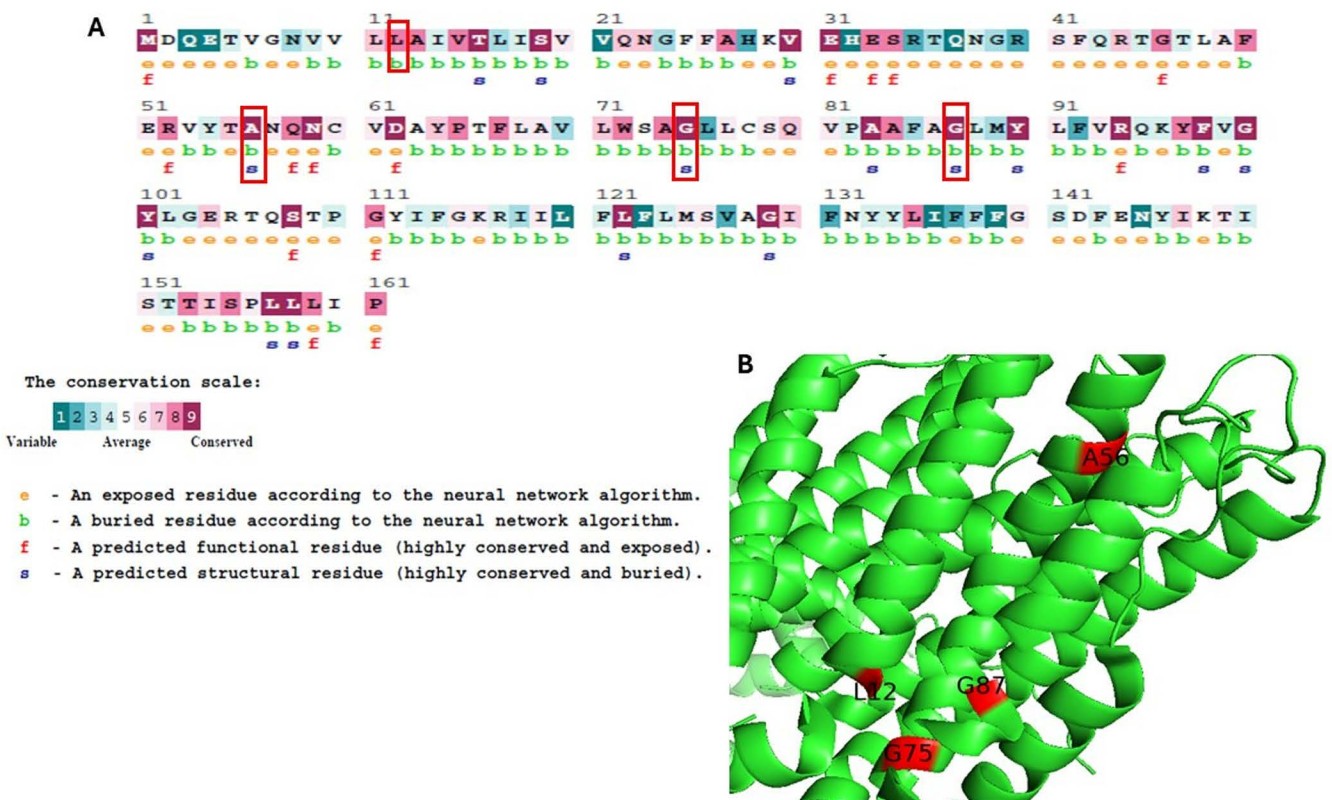

**Fig 2. ConSurf analysis for critical residues in ALOX5AP. (A).** ConSurf analysis results in terms of residue conservation and the 3D of the protein structure. **A.** The amount of confidence in the sequence conservation is shown by a range of colors in the ConSurf results. The sky-blue color in this color scheme stands for variable residues, while the dark purple color stands in for highly conserved residues. **(B)** 3D ALOX5AP indicated the position of the mutants within PDB ID 2Q7M, this figure is prepared using PyMOL [47].

The temperature profiles of the wildtype protein and its mutations displayed slight variations around an average of 303 K, signifying stable thermal conditions during the simulation (S1 Fig). The majority of systems exhibited stable temperature trends, but G75R and G87R demonstrated marginally greater variability over specific intervals. The results indicate that the mutations exerted negligible influence on the overall thermal stability of the ALOX5AP.

The pressure analysis indicated an initial stabilizing period for all systems, succeeded by oscillations around a mean pressure of approximately 0 bar (S2 Fig). Both the wild-type and mutant proteins exhibited comparable patterns, while G75R had marginally more oscillations during equilibration. The data suggest that the mutations exerted little impact on the pressure stability of the ALOX5AP under the simulated conditions. The density analysis indicated that equilibrium was attained at roughly 25 ps for all systems (S3 Fig). The wildtype ALOX5AP and the majority of mutants (A56V, G75R, and G87R) had similar density values (~1010 kg/m³), indicating little structural variations. Nevertheless, the L12F mutant had a somewhat elevated density (~1023 kg/m³) (S3 Fig).

**3.6.2. RMSD and RMSF analyses of ALOX5AP wildtype and its variants.** RMSD profiles of the wildtype ALOX5AP and its mutations (L12F, A56V, G75R, and G87R) during a duration of 100 ns (Fig 3A). The wild-type protein exhibited a persistent RMSD plateau of around 1.2 nm following an initial equilibration phase, signifying consistent conformational dynamics. Among the mutations, L12F demonstrated the lowest RMSD (~0.5 nm). While A56V exhibited greater variability, stabilizing at approximately 1.4 nm. The G75R and G87R mutants exhibited intermediate RMSD values

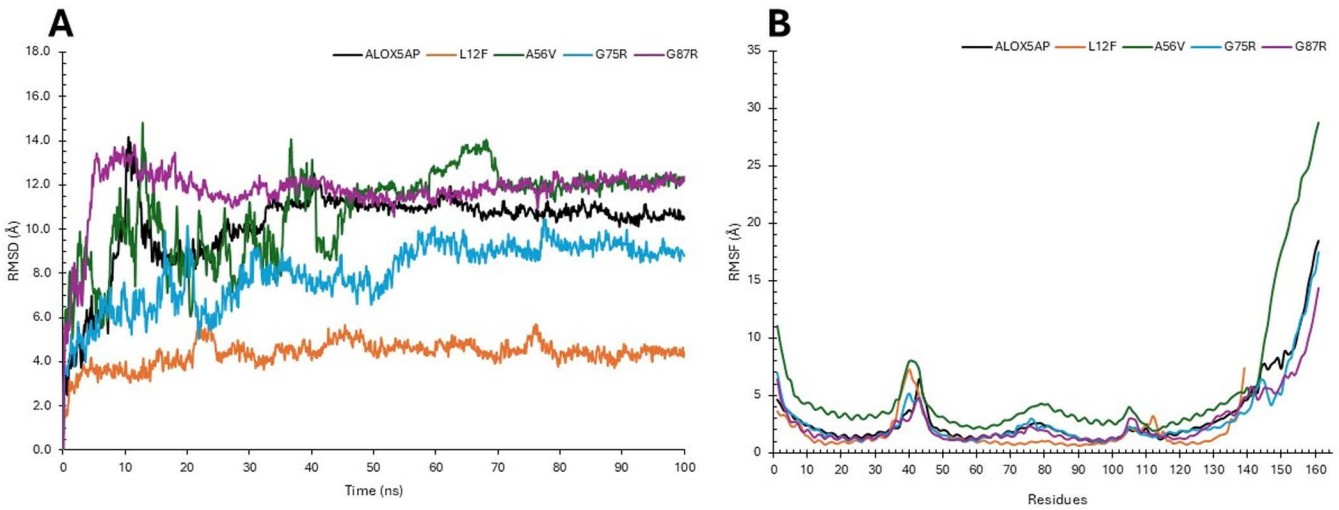

**Fig 3. RMSD (A) and RMSF (B) analyses of ALOX5AP and its Variant proteins.** Comparative MD analyses of wildtype ALOX5AP and selected variants, showing **(A)** RMSD trajectories indicating overall protein stability, and **(B)** RMSF profiles highlighting residue-level flexibility differences (ALOX5AP wildtype in black, L12F in orange, A56V in green, G75R in blue, G87R in purple). The duration of MD simulation was 100 ns.

(~1.0–1.2 nm), similar to the wild-type, although displayed periodic variations, signifying moderate departures in structural stability.

RMSF profiles of the wildtype ALOX5AP and its mutations, emphasizing residue-specific flexibility throughout the simulation duration. The wild-type protein displayed regular changes, with peaks noted around the terminal sections, suggesting inherent flexibility in these places. Comparable tendencies were noted for the mutants, with the terminal areas (residues approaching 160) exhibiting the highest RMSF values (>2.5 nm). The L12F mutant demonstrated the least fluctuations overall, but A56V exhibited heightened variations across the protein, especially in flexible loop regions. G75R and G87R exhibited minor variations, with marginally increased RMSF values near the termini in comparison to the wild-type (Fig 3B).

**3.6.3. Radius of gyration (Rg) Analysis of ALOX5AP Wildtype and its variants.** The Rg patterns for the wildtype ALOX5AP and its mutations (L12F, A56V, G75R, and G87R) during a 100 ns simulation period are shown in Fig 4. The Rg denotes the compactness of the protein structure, with diminished values indicating a more tightly folded conformation. The wildtype ALOX5AP exhibited an average Rg of approximately 2.2 nm during the simulation, with slight variations, signifying a stable and somewhat compact conformation. The L12F mutant had the lowest Rg value (~1.8 nm) with minor variations, signifying a highly compact and stable structure in comparison to the wild-type protein and other mutants. The A56V mutant had the largest Rg (~2.6 nm) among the systems, with considerable oscillations over the simulation duration. The A56V mutation results in a more expansive and flexible structure. The G75R and G87R mutants displayed intermediate Rg values (~2.1–2.3 nm) similar to those of the wildtype protein, with minor variations. The G75R and G87R mutations demonstrate behavior akin to the wildtype protein (Fig 4).

**3.6.4. SASA analysis of ALOX5AP wildtype and its variants.** SASA patterns of the wildtype ALOX5AP and its mutations (L12F, A56V, G75R, and G87R) during a 100 ns simulation period. SASA quantifies the surface area of a protein accessible to the solvent, offering information regarding structural compactness and conformational alterations. The wild-type protein exhibited an average SASA of roughly 115 nm², with slight variations during the simulation, signifying a highly stable structure with uniform solvent exposure. The L12F mutant demonstrated the lowest solvent-accessible surface area (SASA) of around 95 nm², with negligible variability. The A56V mutant had the greatest SASA of around

**Fig 4. Radius of gyration (Rg) Analyses of ALOX5AP wildtype and its Variants.** Comparative analysis of Rg trajectories for Cα atoms, depicting structural compactness and stability differences among wildtype ALOX5AP and mutant variants (ALOX5AP wildtype in black, L12F in orange, A56V in green, G75R in blue, G87R in purple) during MD simulations for 100 ns.

125 nm² among the systems, indicating enhanced solvent exposure. The G75R and G87R mutants demonstrated SASA values akin to the wild-type (~115 nm²), with minor variations during the simulation. The results demonstrate that these alterations do not substantially influence the solvent exposure or compactness of the protein. The SASA analysis demonstrates specific impacts of the mutations on the structural characteristics of the ALOX5AP (Fig 5).

**3.6.5. Hydrogen bond (HB) formation analyses of ALOX5AP wildtype and its variants.** HB is a vital factor in protein stability, structural integrity, and interactions.

The ALOX5AP wildtype exhibited an average of approximately 125 HB during the simulation, with consistent variations, signifying stable internal bonding and structural integrity.

The L12F mutant displayed the fewest HBs (about 100), with diminished variations relative to the wild-type and other mutants. The A56V mutant exhibited a moderately decreased hydrogen bond count (~110) relative to the wild-type, accompanied by somewhat increased fluctuations. The G75R and G87R mutants exhibited HB counts (~115–120) akin to the wildtype, with analogous oscillations. This indicates that these mutations exert negligible influence on the overall hydrogen bonding network and structural integrity of ALOX5AP. The study of HBs elucidates the structural implications of the mutations in *ALOX5AP* (Fig 6).

**3.6.6. Simulation trajectory snapshot.** The MD simulation trajectory of the wildtype ALOX5AP and its variants (L12F, A56V, G75R, and G87R) reveals that most of the protein remains stable, with well-aligned helices and minimal deviations in the core structure (Fig 7). However, the loops are inherently more flexible, which is biologically relevant since loops

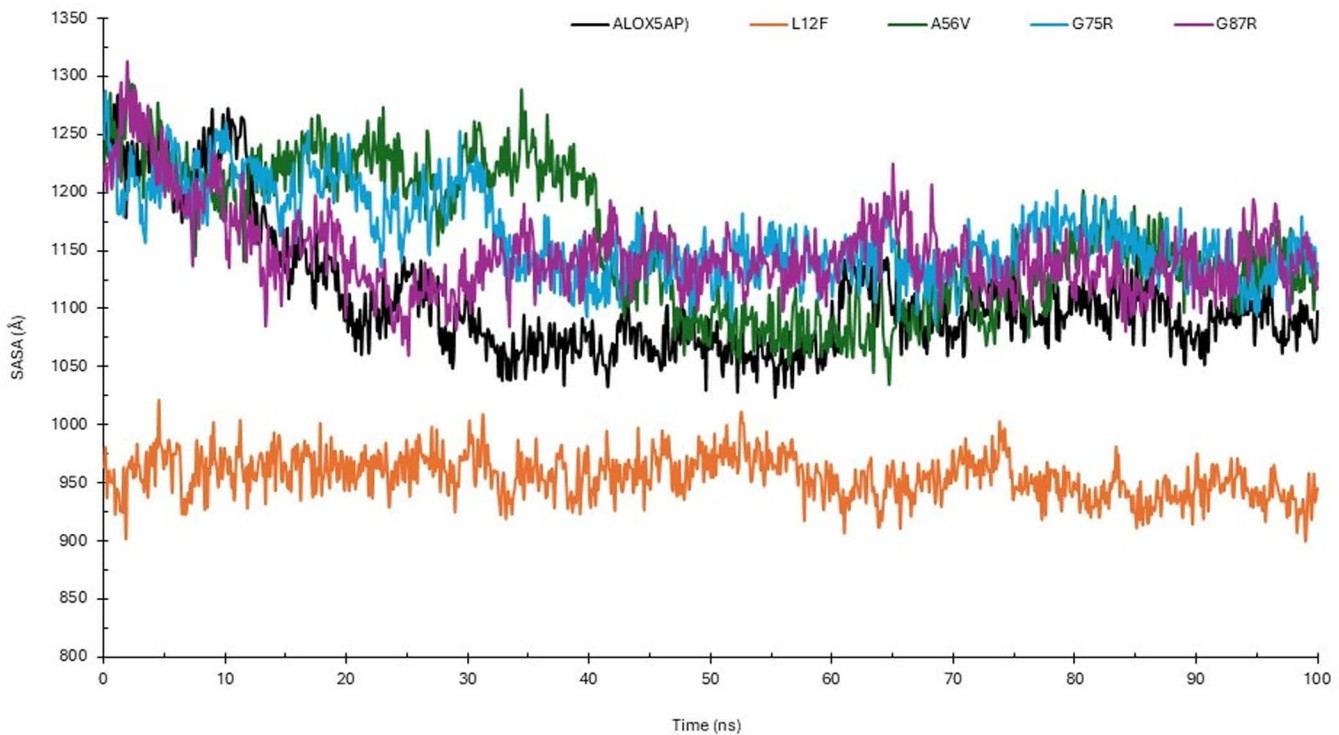

**Fig 5. SASA analyses of ALOX5AP wildtype and its variants.** SASA analyses of ALOX5AP wildtype and its variants for 100 ns. (ALOX5AP wildtype in black, L12F in orange, A56V in green, G75R in blue and G87R in purple).

naturally exhibit greater flexibility. Interestingly, some mutations, such as L12F and A56V, induce notable flexibility in specific loop regions, but this instability is relatively mild compared to the more pronounced effects seen in the G75R and G87R mutants. These latter mutations cause widespread structural perturbations, including disrupted helical packing and increased loop flexibility, likely due to steric or electrostatic changes introduced by the mutations.

## Discussion

Cardiovascular diseases (CVDs) and cerebrovascular diseases are one of the main causes of deaths and disabilities worldwide with 20.5 million deaths in the year 2021 [48,49]. Certain genetic loci were reported to be associated with CVDs with exome and genome-wide sequencing methods [50,51]. Genetic testing is used to identify the individuals and populations at risk for CVDs and other diseases [52,53]. Maintaining a healthy lifestyle such as sound nutrition, regular physical activity, no alcohol, no smoking and weight loss will reduce the risk of CVDs [54].

The ALOX5AP is important for the synthesis of leukotrienes, which is involved in inflammatory reactions and dysregulation of leukotrienes production results in atherosclerosis [55,56]. Atherosclerosis is main pathogenic cause of CAD, cerebral stroke, peripheral arterial disease [57]. Jin et al reported that the SNPs (rs17216473, rs10507391, rs4769874, rs9551963, rs17222814, and rs7222842) of *ALOX5AP* gene were associated with peripheral arterial disease in elderly Korean population [58]. While, Lee et al., reported that *ALOX5AP* rs4293222, rs10507391, rs12429692 SNPs were associated with risk of atherothrombotic stroke in Taiwanese population [59]. Furthermore, it has been demonstrated that the *ALOX5AP* 17216473 SNP is linked to CAD in Ukrainian Chinese populations [60,61]. In addition, rs4073259, rs9579646, rs9551963, rs9315050, rs9551963 and the rs4073259 SNPs of *ALOX5AP* were linked with ischemic stroke in Chinese

Han population [62]. Moreover, the *ALOX5AP* rs38022789 SNP was reported to be associated with diabetic nephropathy in Slovenian population [63]. Whereas, the *ALOX5AP* rs10507391 SNP was reported to increase susceptibility to myeloid leukemia in Chinese population [22].

The protein structure affects its protein-protein interaction and function [64]. SNPs may influence protein structure and function and therefore bioinformatics, protein structural and function studies are required to gain insight on the impact of SNPs on protein [65–68]. In the present study we investigated the effects of nsSNPs the structure of ALOX5AP using bio-informatics tools. We employed MD simulation to examined effect of nsSNPs L12F, A56V, G75R, and G87R on ALOX5AP structural stability since *ALOX5AP* gene variations are associated with CVDs and other diseases [17,63,69,70].

Our results indicated L12F results in exchange of leu 12 to Phe (Tables 1–3, and Fig 3). This variant results in structural changes of ALOX5AP leading to loss of helix, loop, loss of GPI-anchor amidation at N8, transmembrane protein change and alteration of signal peptide (Table 4). Furthermore, results showed that L12F variant exhibited a slightly elevated density (~1023 kg/m³), indicated possible modifications of ALOX5AP in packing and compactness (S3 Fig). Result also showed that the L12F variant may affect the structural integrity of ALOX5AP more profoundly than the other variants. The L12F variant demonstrated the lowest RMSD (~0.5 nm), (Fig 3A), and highest RMSF values (>2.5 nm) (Fig 3B) indicating a more compact and stable structure. The L12F variant showed the least fluctuations overall aligning with its compact structure. RMSD and RMSF analyses demonstrate that the L12F mutation enhances ALOX5AP stability, leading to a more ALOX5AP compact conformation. Moreover, our Rg results indicated that the L12F mutation appears to improve protein folding and stability (Fig 4). Whereas the SASA analysis (Fig 5) showed that L12F variant had diminished solvent exposure providing a more compact structure relative to the ALOX5AP wildtype and other variants, implying that L12F variant augments structural compactness of ALOX5AP. HB data (Fig 6) indicated that the L12F variant diminishes the total hydrogen bonding capability, potentially leading to a more compact and stiff structure as observed in prior assessments. The L12F variant results in an increased ALOX5AP compact conformation (Fig 7) leading to structural changes which

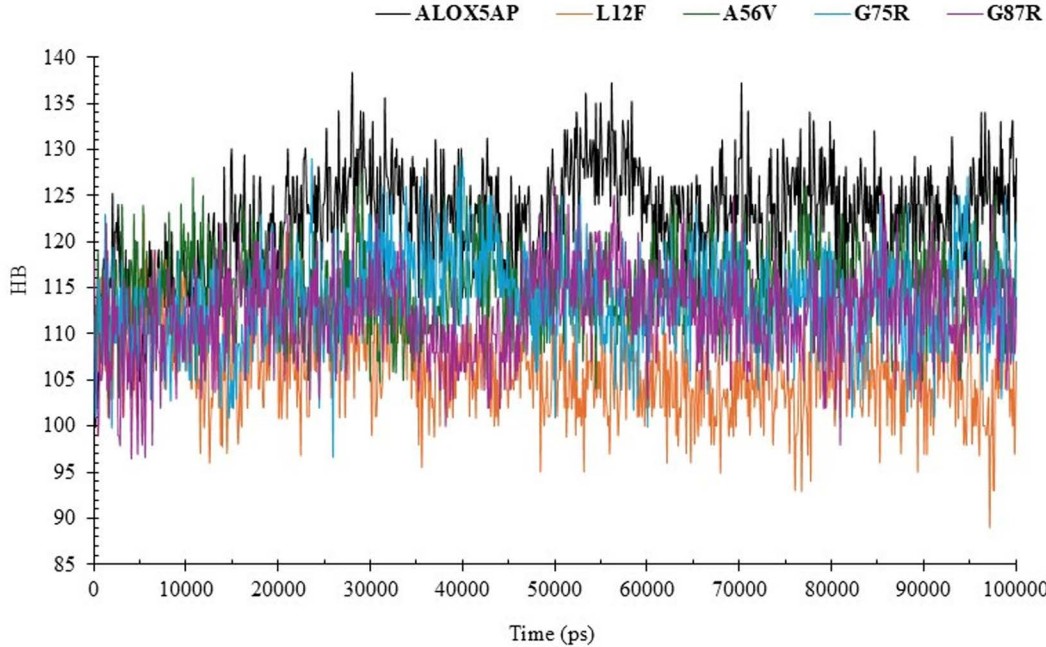

**Fig 6. Hydrogen bond (HB) formation analyses of ALOX5AP wildtype and its variants.** HB formation analyses of ALOX5AP wildtype and its variant proteins for 100 ns (ALOX5AP wildtype in black, L12F in orange, A56V in green, G75R in blue and G87R in purple).

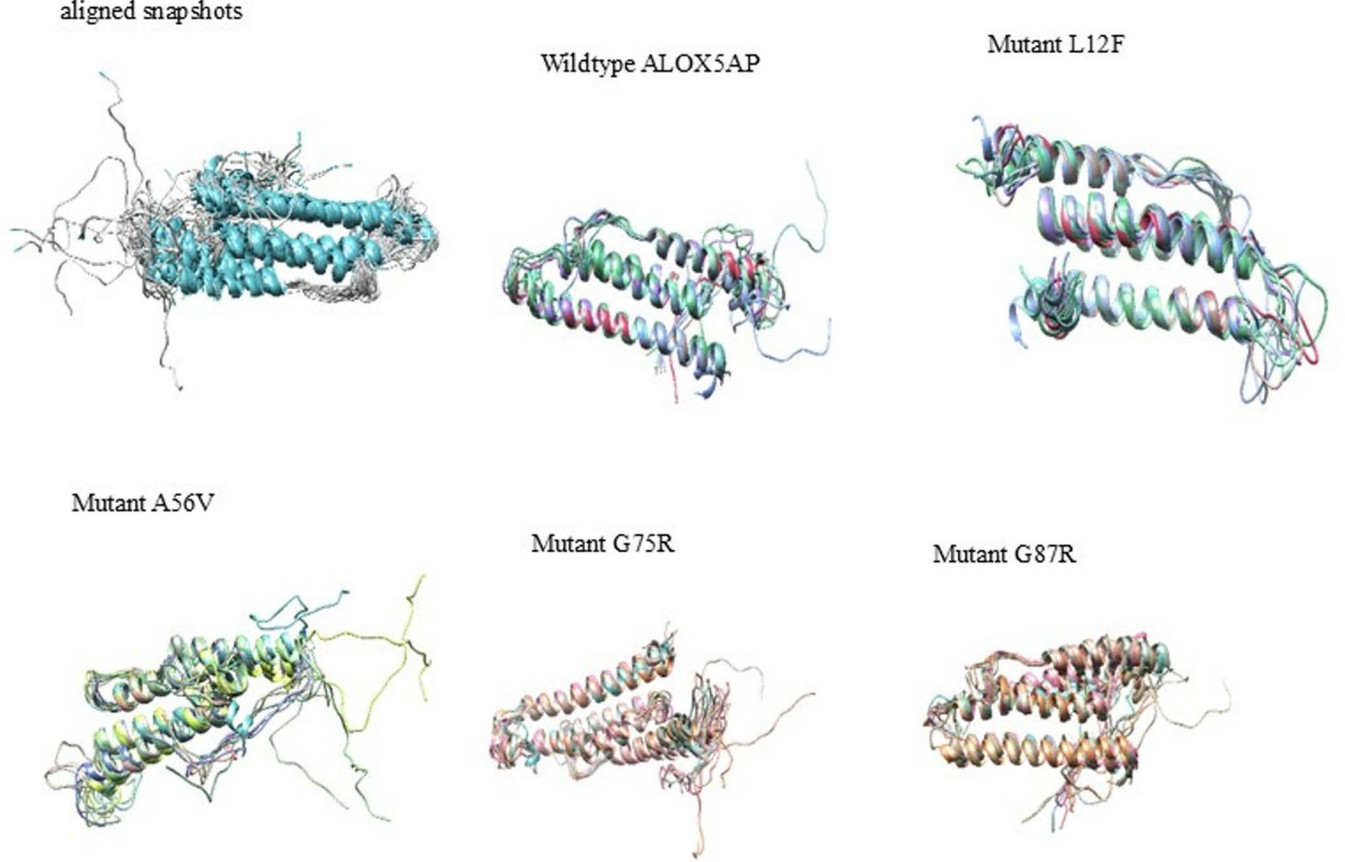

**Fig 7. Simulation trajectory snapshots of ALOX5AP wildtype and its variants.** Structural comparison of ALOX5AP wildtype and its Variant proteins based on simulation trajectory snapshots.

can significantly influence the functional properties of the ALOX5AP. This may be consistent with a study reported that a mutation leading to more compact protein can alter protein structure dynamics and significantly affect the function of the protein [71].

Results showed that the A56V variant leads to ALOX5AP loss of helix and pathogenic (Tables 1–4). Result of density experiment of the A56V variant (~1010 kg/m³) was comparable to that of the ALOX5AP wild type, implying little structural change (S3 Fig). In RMSD experiment, the A56V showed greater variability, stabilizing at about 1.4 nm, indicating enhanced structural flexibility. In the RMSF experiment, the A56V showed the greatest variations across the ALOX5AP, especially in flexible loop regions. In summary, the RMSD and RMSF analyses showed that the A56V mutation promotes the flexibility and protein structural variability, and weakens the stability of the ALOX5AP (Fig 3). Whereas our SASA analysis showed that the A56V variant had the greatest Rg (~2.6 nm) than the ALOX5AP wildtype and its variants (Fig 4). with considerable oscillations over the simulation duration. The A56V mutation leads to a more expansive and flexible structure of ALOX5AP, indicating reduced compactness and possible instability of ALOX5AP. In addition, the SASA run showed that the A56V mutant had the biggest SASA (~125 nm²) than the ALOX5AP wildtype and its variants, indicating enhanced solvent exposure (Fig 5). This indicates that the A56V variant induces a more extended and less compact structure, potentially signifying destabilization or modified folding of ALOX5AP. The HB experiment indicated that A56V mutant exhibited a slightly reduced HB (~110) relative to the ALOX5AP wildtype, agreeing with its increased fluctuations (Fig 6). This

suggests that the A56V mutation may interfere with hydrogen bonding interactions, resulting in diminished stability and structural integrity. The results of the simulation trajectory snapshots indicated that the A56V results in notable flexibility in specific ALOX5AP loop regions (Fig 7).

Result also showed that the G75R and G87R variants influence the ALOX5AP structure (Tables 1–4). These variants result in substitution of glycine 75, 87, respectively (un uncharged amino acid) to arginine (a positively charged) amino acid. Result indicated that G75R and G87R variants result in loss of pyrrolidone carboxylic acid at Q80 of ALOX5AP and altered ordered interface, respectively, of ALOX5AP. Our RMSD and RMSF data showed that the G75R and G87R exhibited minor variations, with marginally increased RMSF values near the termini in comparison to the wild-type (Fig 3). The G75R and G87R variants displayed intermediate Rg values (~2.1–2.3 nm) similar to those of the wildtype protein (Fig 4). These findings indicate that G75R and G87R variant exert a mild influence on the compactness of ALOX5AP structure without causing substantial instability. Result of SASA analysis showed that the G75R and G87R variants exert no influence on the ALOX5AP solvent exposure, preserving characteristics similar to the wildtype (Fig 5). Moreover, G75R and G87R variants preserved HB formation patterns comparable to ALOX5AP wildtype, suggesting negligible effects on the protein's internal bonding and stability (Fig 6). In addition, simulation trajectory snapshots results indicated G75R and G87R variants induce instability causing widespread structural perturbations of ALOX5AP, including disrupted helical packing and increased loop flexibility, likely due to steric or electrostatic changes introduced by Arginine (Fig 7). The G75R and G87R variants potentially impact ALOX5AP dynamics, stability, and function.

The impaired ALOX5AP function due to the mutations would affects the leukotrienes biosynthesis. Leukotrienes are crucial pro-inflammatory lipid mediators, play important roles in the pathogenesis of acute or chronic inflammatory diseases, and implicated in immune-mediated disorders such as bronchial asthma, rhinitis, atherosclerotic CVD and stroke [55,56,72]. Limitations of the present study include that this in silico analysis was performed on certain genetic variants, and other variants of ALOX5AP, potentially critical to clinical outcomes, were not analyzed. Furthermore, the MD simulations do not fully represent the complexity of cellular environments, including intricate interactions with arachidonic acid and 5-lipoxygenase, cell membrane and other physiological conditions which may affect ALOX5AP function.

## Conclusion

To sum up, in this MD simulation study we investigated the effects of certain nsSNPs on the structure of the ALOX5AP. The L12F, A56V, G75R, and G87R using bioinformatics tools. Results showed that variants influence the ALOX5AP differently. L12F variant caused noticeable structural changes, making ALOX5AP more compact and stable. These changes included improved structural integrity, reduced flexibility, and decreased solvent exposure. While A56V variant has a destabilizing effect, increasing ALOX5AP flexibility and reducing its compactness and stability. The G75R and G87R variants, while causing some localized structural disruptions, had relatively minor effects overall and retained much of the stability seen in ALOX5AP wild-type. Our study highlights how these variations in the ALOX5AP gene can differently affect the protein's structure, stability, and function leading to defective leukotrienes biosynthesis, shedding light on their potential contributions to the development of atherosclerotic CVDs and other diseases reported to be associated with ALOX5AP variants. Future MD simulations (to more nsSNPs) in which the complexity of cellular environment is presented, large scale case-control and protein functional studies are required to validate these findings.

## Supporting information

**S1 Fig. Simulation of the temperature changes over time for the ALOX5AP wildtype and its variant proteins.**
(TIF)

**S2 Fig. Simulation of the pressure changes over time for the ALOX5AP wildtype and its variant proteins.**
(TIF)

**S3 Fig. Simulation of the density changes over time for the ALOX5AP wildtype and its variant proteins.**
(TIF)

## Author contributions

**Conceptualization:** Mohamed E. Elnageeb, Imadeldin Elfaki, Gad Allah Modawe, Abdelrahman Osman Elfaki, Othman R Alzahrani, Hytham A. Abuagla, Hayam A. Alwabsi, Adel I. Alalawy, Mohammad Rehan Ajmal, Elsiddig Idriss Mohamed, Hussein Eledum, Syed Khalid Mustafa, Elham M Alhathli.

**Formal analysis:** Mohamed E. Elnageeb, Imadeldin Elfaki, Othman R Alzahrani, Hayam A. Alwabsi, Mohammad Rehan Ajmal, Hussein Eledum, Syed Khalid Mustafa.

**Funding acquisition:** Imadeldin Elfaki.

**Investigation:** Mohamed E. Elnageeb, Imadeldin Elfaki, Gad Allah Modawe, Othman R Alzahrani, Hytham A. Abuagla, Hayam A. Alwabsi.

**Methodology:** Mohamed E. Elnageeb, Imadeldin Elfaki.

**Project administration:** Imadeldin Elfaki.

**Resources:** Hytham A. Abuagla.

**Software:** Mohamed E. Elnageeb, Imadeldin Elfaki, Abdelrahman Osman Elfaki, Othman R Alzahrani, Hytham A. Abuagla.

**Supervision:** Imadeldin Elfaki.

**Validation:** Mohamed E. Elnageeb, Imadeldin Elfaki, Gad Allah Modawe, Hytham A. Abuagla, Adel I. Alalawy, Mohammad Rehan Ajmal, Elsiddig Idriss Mohamed, Hussein Eledum, Elham M Alhathli.

**Visualization:** Mohamed E. Elnageeb, Imadeldin Elfaki, Abdelrahman Osman Elfaki, Hayam A. Alwabsi, Adel I. Alalawy, Mohammad Rehan Ajmal, Elsiddig Idriss Mohamed, Hussein Eledum, Syed Khalid Mustafa, Elham M Alhathli.

**Writing – original draft:** Mohamed E. Elnageeb, Imadeldin Elfaki, Othman R Alzahrani.

**Writing – review & editing:** Imadeldin Elfaki.

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
