## [Decision Letter · Decision Letter 0]

Dear Dr. Elfaki,

Thank you for submitting your manuscript to PLOS ONE. After careful consideration, we feel that it has merit but does not fully meet PLOS ONE’s publication criteria as it currently stands. Therefore, we invite you to submit a revised version of the manuscript that addresses the points raised during the review process.

After carefully considering the reviews and assessing your manuscript, I am pleased to inform you that we would like to invite you to revise and resubmit your manuscript for further consideration. The reviewers have provided constructive comments that will help strengthen your work. Please address each of these points thoroughly in your revised manuscript. Additionally, ensure that you provide a detailed response letter outlining how you have addressed each comment raised by the reviewers. This will help the reviewers and myself to evaluate the changes made to the manuscript.  Please note that additional references suggested during the peer-review process should only be included if the authors agree that they are relevant and useful. Good luck

We look forward to receiving your revised manuscript.

Kind regards,

Khalid Raza, PhD (Computational Biology)

Academic Editor

PLOS ONE

Journal Requirements:

5. We notice that your supplementary figures are uploaded with the file type 'Figure'. Please amend the file type to 'Supporting Information'. Please ensure that each Supporting Information file has a legend listed in the manuscript after the references list.

6. We note you have included a table to which you do not refer in the text of your manuscript. Please ensure that you refer to Table 1, 2, 3, 4, in your text; if accepted, production will need this reference to link the reader to the Table.

Reviewers' comments:

Reviewer's Responses to Questions

**Comments to the Author**

1. Is the manuscript technically sound, and do the data support the conclusions?

Reviewer #1: Partly

Reviewer #2: Yes

2. Has the statistical analysis been performed appropriately and rigorously?

Reviewer #1: No

Reviewer #2: N/A

3. Have the authors made all data underlying the findings in their manuscript fully available?

Reviewer #1: Yes

Reviewer #2: Yes

4. Is the manuscript presented in an intelligible fashion and written in standard English?

Reviewer #1: No

Reviewer #2: Yes

Reviewer #1: The manuscript contains numerous grammatical errors and typographical mistakes throughout. A thorough revision is necessary before resubmission. Editing by a native English speaker or professional language editing service is strongly recommended.

Section 2.2: Data Collection

References are missing. Please provide appropriate citations to support the methodology described.

Section 2.5: Molecular Dynamics (MD) Simulations Using GROMACS

The section lacks sufficient citations. Key methodological steps and tools used in the simulations should be properly referenced.

Section 3.1: Prediction of Deleterious nsSNPs in ALOX5AP

The classification of mutations is not adequately referenced. For example, the manuscript states that SIFT classified all examined nsSNPs as harmful, with scores between 0 and 0.03. However, the rationale for selecting this threshold is not provided or supported with references.

Throughout this section, several predictive tool results (e.g., SNPs&GO, PROVEAN, PhD-SNPg) are reported without proper citation or explanation of the basis of score ranges used for classification. This reduces the credibility of the findings and should be addressed.

Discussion

The discussion lacks scientific depth and critical comparison with findings from other published studies.

The opening part of the discussion reads more like an introduction and should be revised to focus on interpreting results.

The impact of the identified mutations, particularly in relation to cardiovascular diseases, needs to be discussed in more detail. For example, although the figures suggest that the L12F variant causes significant structural compaction of ALOX5AP, the manuscript fails to explain how this structural change could affect protein function or contribute to cardiovascular disease mechanisms.

The sentence: “Our result in inconsistent with a study reported that the p.L12F variant has no detrimental effect on the pancreatic secretory trypsin inhibitor protein(47), this inconsistency may because of the different protein examined in that study(47).” is problematic both in grammar and logic. If the compared study involves an entirely different protein, then drawing a direct comparison based on the same amino acid substitution (L12F) is inappropriate due to differences in tertiary structure and protein context. This kind of comparison is premature and scientifically unsound.

Conclusion

The conclusion section would benefit from a brief discussion of the study’s limitations. Additionally, future research directions should be clearly outlined to enhance the manuscript’s overall impact.

Reviewer #2: Dear authors,

Your manuscript is somehow interesting. However, the major issue which comes in my mind, plot the simulation results in in Angstrom, not in nm to make it clear how much the results has deviated. Also, compare it with standard results.

**Do you want your identity to be public for this peer review?** For information about this choice, including consent withdrawal, please see our Privacy Policy

Reviewer #1: **Yes: ** Dr Ayan Saha

Assistant Professor

Bioinformatics and Biotechnology

Asian University for Women

Chattogram-4000, Bangladesh.

Reviewer #2: No

---

## [Author Response · Author response to Decision Letter 1]

11 Jun 2025

Dear Editor, Dear reviewers

Thank you very much for your consideration. We also thank you and the reviewers for the questions and the comments that significantly improved our manuscript. Please find below step-by-step response to all questions and comments.

Best regards,

Imadeldin Elfaki, PhD

The manuscript contains numerous grammatical errors and typographical mistakes throughout. A thorough revision is necessary before resubmission. Editing by a native English speaker or professional language editing service is strongly recommended.

Authors

Thank you very much. We are very sorry that manuscript contains numerous grammatical errors and typographical mistakes. English is revised carefully.

Section 2.2: Data Collection

● References are missing. Please provide appropriate citations to support the methodology described.

Authors

● Thank you very much. Appropriate citations to support the methodology described are provided. Please see ref 24 to 45.

Section 2.5: Molecular Dynamics (MD) Simulations Using GROMACS

● The section lacks sufficient citations. Key methodological steps and tools used in the simulations should be properly referenced.

● Authors

● Thank you very much. In molecular dynamics (MD) simulation the temperature, pressure, and density [1, 2] characteristics of the wild-type ALOX5AP protein and its mutants (L12F, A56V, G75R, and G87R) were examined throughout a 100 ps simulation period to evaluate the structural and dynamic impacts of these variants. to examine the structural stability and flexibility of ALOX5AP key metrics such as the root-mean-square deviation (RMSD) [1, 2], which assesses the ALOX5AP overall structural stability, and root-mean-square fluctuation (RMSF) [1, 2], which evaluated the flexibility of individual amino acid residue. We also examine the radius of gyration (Rg) which gives an understanding of the overall dimensions of ALOX5AP, and solvent- accessible surface area (SASA) that is a bio-molecular surface area accessibility to solvent molecules [1, 3]. Hydrogen bonds analysis was also examined, HB is an important factor in protein stability, structural integrity, and interactions[4]. To enhance the data interpretation, custom Python scripts were developed [5] to generate visualizations of these parameters, enabling a comprehensive understanding of the simulation results.

●

Section 3.1: Prediction of Deleterious nsSNPs in ALOX5AP

● The classification of mutations is not adequately referenced. For example, the manuscript states that SIFT classified all examined nsSNPs as harmful, with scores between 0 and 0.03. However, the rationale for selecting this threshold is not provided or supported with references.

● Authors

● Thank you very much. In SIFT Score, the amino acid substitution is predicted damaging if the score is <= 0.05[6], please also see website https://sift.bii.a-star.edu.sg/www/SIFT_help.html. The SIFT Score of the selected variants L12F, A56V, G75R and G87R variants was 0 (Table1). This score is predicted damaging or deleterious to the protein.

● Throughout this section, several predictive tool results (e.g., SNPs&GO, PROVEAN, PhD-SNPg) are reported without proper citation or explanation of the basis of score ranges used for classification. This reduces the credibility of the findings and should be addressed.

● Authors

● -Thank you very much. In the SNPs&Go, score of 0.5 is selected to distinguish between benign (t≤0.5) and pathogenic (t>0.5) SNP[7]. SNPs&GO categorized all four examined nsSNPs as "pathogenic," with pathogenicity probability (Path_Prop) between 0.584 and 0.953[7] (Table2).

● -We did not employ PROVEAN in this study.

● - The PhD-SNP classified all four variations as "disease-associated," with reliability indices (RI) between 2 and 9, indicating the confidence in the predictions[8].

Discussion

● The discussion lacks scientific depth and critical comparison with findings from other published studies.

● Authors

● Thank you very much. A comparison with findings from other published studies was added in a separated paragraph. Changes are highlighted.

● The opening part of the discussion reads more like an introduction and should be revised to focus on interpreting results.

Authors

opening part of the is revised and focused on interpreting results

● The impact of the identified mutations, particularly in relation to cardiovascular diseases, needs to be discussed in more detail. For example, although the figures suggest that the L12F variant causes significant structural compaction of ALOX5AP, the manuscript fails to explain how this structural change could affect protein function or contribute to cardiovascular disease mechanisms.

● Authors

● Thank you very much. We explained how this structural change due to the mutation could affect ALOX5AP function or contribute to cardiovascular disease mechanisms.

● The sentence: “Our result in inconsistent with a study reported that the p.L12F variant has no detrimental effect on the pancreatic secretory trypsin inhibitor protein(47), this inconsistency may because of the different protein examined in that study(47).” is problematic both in grammar and logic. If the compared study involves an entirely different protein, then drawing a direct comparison based on the same amino acid substitution (L12F) is inappropriate due to differences in tertiary structure and protein context. This kind of comparison is premature and scientifically unsound.

● authors

● Thank you very much. We remove the sentence “Our result in inconsistent with a study reported that the p.L12F variant has no detrimental effect on the pancreatic secretory trypsin inhibitor protein.

Conclusion

● The conclusion section would benefit from a brief discussion of the study’s limitations. Additionally, future research directions should be clearly outlined to enhance the manuscript’s overall impact.

● Authors

● Thank you very much. Some study limitations and suggested future research direction are added to the conclusion.

REFERENCES

1. Hasnain MJU, Shoaib M, Qadri S, Afzal B, Anwar T, Abbas SH, Sarwar A, Talha Malik HM, Tariq Pervez M: Computational analysis of functional single nucleotide polymorphisms associated with SLC26A4 gene. PLoS One 2020, 15(1):e0225368.

2. Alomair L, Mustafa S, Jafri MS, Alharbi W, Aljouie A, Almsned F, Alawad M, Bokhari YA, Rashid M: Molecular Dynamics Simulations to Decipher the Role of Phosphorylation of SARS-CoV-2 Nonstructural Proteins (nsps) in Viral Replication. Viruses 2022, 14(11).

3. Rani N, Boora N, Rani R, Kumar V, Ahalawat N: Molecular dynamics simulation of RAC1 protein and its de novo variants related to developmental disorders. J Biomol Struct Dyn 2024, 42(24):13437-13446.

4. Zhang D, Lazim R: Application of conventional molecular dynamics simulation in evaluating the stability of apomyoglobin in urea solution. Sci Rep 2017, 7:44651.

5. Irrgang ME, Davis C, Kasson PM: gmxapi: A GROMACS-native Python interface for molecular dynamics with ensemble and plugin support. PLoS Comput Biol 2022, 18(2):e1009835.

6. Ng PC, Henikoff S: Predicting deleterious amino acid substitutions. Genome Res 2001, 11(5):863-874.

7. Capriotti E, Martelli PL, Fariselli P, Casadio R: Blind prediction of deleterious amino acid variations with SNPs&GO. Hum Mutat 2017, 38(9):1064-1071.

8. Capriotti E, Calabrese R, Casadio R: Predicting the insurgence of human genetic diseases associated to single point protein mutations with support vector machines and evolutionary information. Bioinformatics 2006, 22(22):2729-2734.

---

## [Decision Letter · Decision Letter 1]

Dear Dr. Elfaki,

Thank you for submitting your manuscript to PLOS ONE. After careful consideration, we feel that it has merit but does not fully meet PLOS ONE’s publication criteria as it currently stands. Therefore, we invite you to submit a revised version of the manuscript that addresses the points raised during the review process.

**Before manuscript may be accepted, authors are required to address the following concerns:**
**1) Drop the word "(MD)" from the title.**
**2) Section 2.1 heading, check the spelling of "Plane". **
**3) A large number of abbreviations have been used. Compile a list of abbreviations used and place it before the References section**
**4) There spacing issues and other typos throughout the manuscript. For instance, there is no space between Figure and figure numbers. Get your manuscript thoroughly proofread.**

We look forward to receiving your revised manuscript.

Kind regards,

**Dr. Khalid Raza, PhD (Computational Biology)**

*Academic Editor*

PLOS ONE

**Journal Requirements:**

Reviewers' comments:

Reviewer's Responses to Questions

**Comments to the Author**

Reviewer #1: All comments have been addressed

Reviewer #2: All comments have been addressed

2. Is the manuscript technically sound, and do the data support the conclusions?

Reviewer #1: Partly

Reviewer #2: Yes

3. Has the statistical analysis been performed appropriately and rigorously?

Reviewer #1: N/A

Reviewer #2: Yes

4. Have the authors made all data underlying the findings in their manuscript fully available?

Reviewer #1: No

Reviewer #2: Yes

5. Is the manuscript presented in an intelligible fashion and written in standard English?

Reviewer #1: Yes

Reviewer #2: Yes

**Reviewer #1:**  The manuscript is in good form for acceptance, but kindly review the grammar and typos once again. Improve the quality of figures also.

**Reviewer #2: ** The manuscript is now well improved as the authors has addressed all of my comments and can be accepted for publication.

**Do you want your identity to be public for this peer review?** For information about this choice, including consent withdrawal, please see our Privacy Policy

Reviewer #1: **Yes: ** Dr Ayan Saha

Reviewer #2: **Yes: ** Shaban Ahamd

---

## [Author Response · Author response to Decision Letter 2]

7 Jul 2025

Dear Editor

Thank you very much for the considering our paper. We also thank you and the reviewers the comments that significantly improved our manuscript. Please find below responses to questions and comments.

Best regards,

Imadeldin Elfaki, PhD

1) Drop the word "(MD)" from the title.

Authors

Thank you very much. MD is dropped from the title.

2) Section 2.1 heading, check the spelling of "Plane".

Authors

Thank you very much: The spelling of ‘’plane’’ is checked.

3) A large number of abbreviations have been used. Compile a list of abbreviations used and placed it before the References section

Authors

Thank you very much. A list of abbreviations used is prepared and placed before the References section

4) There spacing issues and other typos throughout the manuscript. For instance, there is no space between Figure and figure numbers. Get your manuscript thoroughly proofread.

Authors

Thank you very much. Types errors are revised; the manuscript is thoroughly proofread.

---

## [Editor Report · Decision Letter 2]

Computational Prediction of the Pathogenic Variants of Arachidonate 5-Lipoxygenase Activating Protein Using Molecular Dynamics Simulation

PONE-D-25-17796R2

Dear Dr. Elfaki,

We’re pleased to inform you that your manuscript has been judged scientifically suitable for publication and will be formally accepted for publication once it meets all outstanding technical requirements.

Kind regards,

Khalid Raza, PhD (Computational Biology)

Academic Editor

PLOS ONE
---

## [Editor Report · Acceptance letter]

PONE-D-25-17796R2

PLOS ONE

Dear Dr. Elfaki,

I'm pleased to inform you that your manuscript has been deemed suitable for publication in PLOS ONE. Congratulations! Your manuscript is now being handed over to our production team.

Kind regards,

on behalf of

Dr. Khalid Raza

Academic Editor

PLOS ONE